# Fixing What Fine-Tuning Breaks: A Simple and Efficient Method to Improve Safety Post Domain Adaptation

## Abstract

Safety-aligned language models suffer from a reduction in safety post-finetuning even on benign data. Prior works have highlighted a solution to the issue via further preference optimization in the fine-tuned models; however, this method is computationally expensive and requires domain-specific preference optimization data. In this paper, we aim to alleviate the degradation in the general safety of the fine-tuned language models via a weight shifting methodology, which is both computationally inexpensive, efficient, and does not require in-domain preference optimization data. We further demonstrate that our methodology has statistically insignificant changes to the model's general coherence and false refusal rates and retains the model's domain-specific knowledge. Finally, we discovered that our method also increases the domain-specific safety of the language model without requiring domain-specific safety data.

## 1 Introduction

Modern transformer-based large language models (Touvron et al., 2023a; Grattafiori et al., 2024; Team et al., 2024; 2023) undergo safety alignment prior to deployment to ensure safety and trustworthiness (Rafailov et al., 2023; Ethayarajh et al., 2024; Bai et al., 2022). Recent works have shown that language model alignment leads to a significant increase in various dimensions of model safety (Alami et al., 2024; Zhang et al., 2025).

Concurrently, research has been focused on adapting these aligned language models to critical domains like medicine, law, finance etc (Cheng et al., 2024; Anisuzzaman et al., 2025). While research in this direction has seen success in fine-tuning aligned language models efficiently, with the models enjoying competitive domain-specific performance, the downstream effects of such fine-tuning on the safety, trustworthiness, and robustness of the model have been a point of concern in the community (Fraser et al., 2025; Qi et al., 2024b).

More specifically, prior work has shown that the safety of fine-tuned aligned models is extremely volatile, to the extent that even a small amount of adversarial data in the training dataset can lead to a significant decrease in safety. More concerningly, fine-tuning aligned models with completely benign data has also been shown to cause degradation of the model's safety (Qi et al., 2024b; Fraser et al., 2025).

To address such concerns, works have proposed methods of realigning the fine-tuned language models with methods such as preference optimization and reinforcement learning with human feedback (Han et al., 2024). While successful, these methods require significant domain-specific alignment data, which can cause a significant barrier to entry for model alignment in esoteric domains, in addition to being computationally expensive.

Additionally, research in the field of mechanistic interpretability has utilized steering vectors to elicit specific behaviors in language models (Arditi et al., 2024; O'Brien et al., 2025; Lee et al., 2025). Primarily, these steering vectors are injected during inference time (Lee et al., 2025) and are either calculated via methods such as difference in means (Arditi et al., 2024; Belrose, 2023; Marks & Tegmark, 2023; Panickssery et al., 2023) or features in the latent space of a sparse autoencoder (O'Brien et al., 2024; 2025; Cunningham et al., 2023).

In this work, we take inspiration from work in mechanistic interpretability and aim to elicit safety behaviors in fine-tuned language models, which suffer from a degradation of safety post fine-tuning. However, unlike prior work, we inject the steering vectors directly into the weights of the fine-tuned language model by performing a low-rank projection to minimize the downsides of the injection. Our contributions are as follows:

1. Introduce a low-rank weight steering methodology called SPECTRA, which greatly improves the safety of the fine-tuned language models.

2. SPECTRA maintains both domain-specific and general coherence of the language model while aiding safety.

3. SPECTRA improves the domain specific safety of the fine-tuned language model without the need for domain-specific safety data or gradient calculations.

4. We measure the impact of SPECTRA on false refusals (a common downside of activation steering) find that the effects of SPECTRA are statistically insignificant.

5. We show that steering vectors can be utilized to elicit behaviors across models. More specifically, we show that steering vectors in the base model can be used to elicit behaviors in fine-tuned models.

## 2 BACKGROUND

**Transformer:** We utilize a decoder-only Transformer framework (Radford et al., 2019; Vaswani et al., 2017) to transform a sequence of input token indices $\mathbf{t} = (t_1, \ldots, t_n) \in \mathcal{V}^n$ into a sequence of output probability vectors $\mathbf{y} \in \mathbb{R}^{n \times |\mathcal{V}|}$. We define the hidden state $\mathbf{x}_i^{(l)} \in \mathbb{R}^{d_{\text{model}}}$ as the activation within the residual stream for token $i$ at the input of layer $l$. Initialization occurs via the embedding layer such that $\mathbf{x}_i^{(1)} = \texttt{Embed}(t_i)$. This representation evolves through $L$ layers, where each layer applies an attention mechanism followed by a multi-layer perceptron (MLP), both utilizing residual connections:

$$\mathbf{h}_i^{(l)} = \mathbf{x}_i^{(l)} + \texttt{Attn}^{(l)}(\mathbf{x}_{1:i}^{(l)}),$$
$$\mathbf{x}_i^{(l+1)} = \mathbf{h}_i^{(l)} + \texttt{MLP}^{(l)}(\mathbf{h}_i^{(l)}).$$

To generate predictions, the final layer activations are projected back to the vocabulary space via $\texttt{logits}_i = \texttt{Unembed}(\mathbf{x}_i^{(L+1)})$, yielding the final distribution $\mathbf{y}_i = \texttt{softmax}(\texttt{logits}_i)$ (Arditi et al., 2024).

### 2.1 REFUSAL DIRECTION

**Isolation of Refusal Direction** To characterize the subspace responsible for refusal, we employ the *difference-in-means* approach (Belrose, 2023; Marks & Tegmark, 2023). Following Arditi et al. (2024), we compute the centroids of activations for harmful and harmless prompts within the training set. For a given layer $l$ and token position $i$, we define the class-conditional mean activations as:

$$\boldsymbol{\mu}_i^{(l)} = \mathbb{E}_{\mathbf{t} \sim \mathcal{D}_{\text{harmful}}^{(\text{train})}} \left[ \mathbf{x}_i^{(l)}(\mathbf{t}) \right], \tag{1}$$

$$\boldsymbol{\nu}_i^{(l)} = \mathbb{E}_{\mathbf{t} \sim \mathcal{D}_{\text{harmless}}^{(\text{train})}} \left[ \mathbf{x}_i^{(l)}(\mathbf{t}) \right]. \tag{2}$$

The candidate refusal vector is subsequently defined as the difference between these centroids: $\mathbf{r}_i^{(l)} = \boldsymbol{\mu}_i^{(l)} - \boldsymbol{\nu}_i^{(l)}$.

**Optimization and Selection:** The procedure yields a tensor of candidate vectors across all layers $l \in [L]$ and positions $i \in I$. To identify the optimal direction $\mathbf{r} := \mathbf{r}_{i^*}^{(l^*)}$, we evaluate the candidates against a validation set. We select the indices $(i^*, l^*)$ that maximize the vector's causal intervention effect: specifically, the ability to suppress refusal when ablated from harmful prompts in $\mathcal{D}_{\text{harmful}}^{(\text{val})}$ and, conversely, to trigger refusal when injected into harmless prompts in $\mathcal{D}_{\text{harmless}}^{(\text{val})}$. We refer to the normalized version of this optimal vector as $\hat{\mathbf{r}}$.

Table 1: The differences between three types of ASR in our safety evaluation.

# 3 EXPERIMENTAL SETUP

**Models:**    This study focuses on widely used safety-aligned large language models (LLMs) . The selected models are: Llama-2-chat-7b (Touvron et al., 2023b) and Gemma2-9b-it Team et al. (2024).

**Fine-Tuning:**    We selected publicly available fine-tuned variants of Llama2-7b-chat(Touvron et al., 2023b) and Gemma2-9b-it (Team et al., 2024). For Llama2-chat-7b we chose models in medical (Rohanian et al., 2024), law (Cheng et al., 2024) and finance domains (Cheng et al., 2024). For Gemma2-9b-it we chose models in medical (OpenMeditron, 2024) and finance (Abdullah Bezir, 2025) domains.

**Measuring General Coherance:**    We measure the performance of the models by measuring their zero-shot accuracy on 5 tasks from EleutherAI's LM Harness (Gao et al., 2023): HellaSwag (Zellers et al., 2019), BoolQ (Clark et al., 2019), RTE (Wang et al., 2019), ARC Challenge (Clark et al., 2018) and Winogrande (Sakaguchi et al., 2021).

**Measuring Domain Specific Coherence :**    To measure domain-specific coherence, we utilize the various benchmarks for each domain. For medical domain, we chose MedMCQA (Pal et al., 2022) and PubMedQA (Jin et al., 2019). For finance domain we chose FinanceBench (Islam et al., 2023). For law we chose LegalBench (Guha et al., 2023).

For our LegalBench (Guha et al., 2023) considerations, we evaluate on the Abercrombie and Hearsay Tasks and report our findings on these benchmarks.

**Measuring General Safety:**    We measure the safety of the models by evaluating its attack success rate (ASR)[1] in response to harmful instructions. Specifically, we prompt the model using ADVBENCH-EVAL, the first 100 prompts from ADVBENCH, and collect its responses. Following Zou et al. (2023b), we consider an attack as successful if the model's response lacks key patterns indicative of refusal. The ASR is then computed as the ratio of successfully attacked prompts to the total number of prompts evaluated. Following Wei et al. (2024), our safety evaluation considers two use cases: the ASR under non-malicious conditions ($\text{ASR}_{\text{Vanilla}}$), and the ASR under a malicious setting – $\text{ASR}_{\text{Adv-Decoding}}$ Huang et al. (2024b), where the attacker manipulates the decoding process. For $\text{ASR}_{\text{Adv-Decoding}}$, we present results with and without the [INST] wrapper[2]. For consistency, we call them $\text{ASR}^{I}_{\text{Adv-Decoding}}$ and $\text{ASR}^{\times}_{\text{Adv-Decoding}}$, respectively.

**Measuring Domain Specific Safety:**    For the **medical** domain, we calculate the ASR scores on the MedSafeEval (Han et al., 2024) and record our findings. We prompt the models and utilize the harmbench classifier (Mazeika et al., 2024) to evaluate the model's output.

We were unable to find reliable benchmarks for safety evaluations of the legal and finance domains. We created two datasets: **FinSafeEval** and **LawSafeEval**, with 142 and 201 safety-related prompts, respectively. These datasets were formulated via prompting Gemini 2.5 Pro (Comanici et al., 2025) to create a dataset of 2000 safety-related prompts for each domain, while domain-specific safe prompts were provided as a reference. After this, manual labeling was performed to select the best-fit prompts for each domain.

**Measuring Robustness Against Jailbreaks:**    We measure the efficacy of the model's defenses against jailbreak methodologies such as GCG (Zou et al., 2023b), GPTFuzz (Yu et al., 2023), and TAP Mehrotra et al. (2024). Although not commonly used as a measure of a model's overall safety, we aim to understand whether our proposed method can increase the overall robustness of a language model against jailbreaks.

---

[1]Sometimes, we refer to ASR as attack score.

[2]As GEMMA2-9B-IT doesn't have [INST] wrapper, we utilize the standard prompt template provided by Team et al. (2024)

**Datasets for Finding Refusal Directions:** Following Arditi et al. (2024), we construct $\mathcal{D}_{\text{harmful}}$ as a collection of harmful instructions from ADVBENCH (Zou et al., 2023a), MALICOUSINSTRUCT (Huang et al., 2024b), TDC2023 (Mazeika et al., 2024), and HARMBENCH (Mazeika et al., 2024). As for $\mathcal{D}_{\text{harmless}}$, we collect a set of harmless instructions from ALPACA (Taori et al., 2023). Each $\mathcal{D}_{\text{harmful}}$ and $\mathcal{D}_{\text{harmless}}$ includes 160 samples which will be split into train and validation splits of 128 and 32 samples, respectively.

**Datasets for Calculating Low-Rank Matrices:** Following Wei et al. (2024), we use the ALIGN dataset to isolate safety critical ranks in the weight matrices of the fine-tuned model. As shown in (Wei et al., 2024), utilizing this dataset aids in isolating safety-critical ranks in models.

**Evaluation of Refusal:** We follow the prior literature (Lermen et al., 2023; Liu et al., 2024; Robey et al., 2023; Shah et al., 2023; Xu et al., 2023; Zou et al., 2023b) and utilize `Harmbench Llama2-13b Classifier` (Mazeika et al., 2024) to classify outputs as refusal (refusal-score = 1) or successful attack (refusal-score= 0). For the case of jailbreaks such as TAP (Mehrotra et al., 2024), GPTFuzz (Yu et al., 2023) we utilize GPT 3.5 Turbo (OpenAI, 2023) for measuring refusal scores.

## 4 METHODOLOGY

We aim to induce the refusal vector in the fine-tuned models to aid the safety of the language model. For this, we propose a weight orthogonalization scheme via which we directly inject the refusal vector into a specific layer in the language model.

Assume that $W$ is a weight matrix of a specific layer in language model (output projection matrix in the attention layer or MLP projection weight), and $X_{\text{in}}$ is the input of this projection. More precisely, each column of $X_{\text{in}}$ is an activation of a response token before projection $W$. To create $X_{\text{in}} \in \mathbb{R}^{d_{\text{in}} \times n}$, we feed the model with several harmful prompts (e.g., we used ALIGN (Wei et al., 2024) in our experiment) and record the activations for the response tokens in $X_{\text{in}}$. Then, we create a low-rank version of $W$ that its columns are aligned with the activation space of harmful data. Specifically, we want a low rank version of the weight matrix in which the ranks are contributing significantly to the safety of the model; thus, we utilize activation-aware SVD.

**ActSVD**: Following Wei et al. (2024), we store all the response activations before the layer with weight matrix $W$ of rank $\rho$ into $X_{\text{in}} \in \mathbb{R}^{d_{\text{in}} \times n}$ and aim to find a low-rank matrix $\tilde{W}$ such that the Frobenius norm of the change to the output is minimized:

$$\tilde{W} = \arg \min_{\text{rank } \tilde{W} \leq \rho} \|WX_{\text{in}} - \tilde{W}X_{\text{in}}\|_F^2.$$

This is be done by performing SVD on $WX_{\text{in}} \in \mathbb{R}^{d_{\text{out}} \times n}$:

$$USV^\top \approx WX_{\text{in}},$$

where $U \in \mathbb{R}^{d_{\text{out}} \times \rho}$ is the orthogonal matrix corresponding to the top $\rho$ left singular vectors. The minimizer is given by,

$$\tilde{W} = UU^\top W,$$

where $\Pi = UU^\top$ is the orthogonal projection onto the $\rho$ most significant left singular subspace (Wei et al., 2024; Hsu et al., 2021; Yuan et al., 2023).

**SPECTRA**: After calculating $\tilde{W}$, we perform the following weight orthogonalization trick:

$$W' = W + \alpha r r^T \tilde{W} \tag{3}$$

$W$ is generally the attention output or MLP projection weight, and $r$ is the refusal direction under consideration with $\alpha$ being the steering hyperparameter.

**Theorem 1.** *Let $x \in \mathbb{R}^{d_{in}}$ be an activation for a specific token, and $h = Wx \in \mathbb{R}^{d_{out}}$. Then, weight steering introduce in equation 3 is equivalent to activation steering defined by $h' = h + \beta \cdot r$, where $\beta$ is a scalar and depends on $x$.*

*Proof.* In the new model where $W$ is replaced by $W'$, we have,

$$h' = W' \cdot h = Wx + \alpha rr^T UU^T Wx = h + (\alpha \cdot r^T UU^T h)r = h + \beta \cdot r, \quad (4)$$

where $\beta = \alpha \cdot r^T UU^T h$ is a scalar. □

The above theorem implies that if $h$ is strongly aligned with the space defined by $U$, then $W'$ implies strong activation steering toward refusal direction $r$. On the other hand, if $h$ is not in the subspace defined by $U$, $\beta$ effectively would be zero and leads to no activation steering. Intuitively, we expect $h$ is strongly aligned with the space defined by $U$, if the original prompt is harmful since $U$ is calculated based on harmful prompts (i.e., $X_{\text{in}}$).

## 5 RESULTS

### 5.1 BEST PRACTICES FOR SPECTRA

We firstly aim to elucidate the optimal method for injecting the refusal vector into the weights of the fine-tuned model. For this experiment, we inject the refusal vector directly into the projection weights of the language models without any low-rank approximation to assess which weights and refusal vectors are most suitable for weight injection. The injection takes the following form:

$$W' = W + \alpha rr^T W \quad (5)$$

We also note that equation 5 is a special case of our method and has been studied under a different setting by Chhabra & Khalili (2025). The refusal vectors and weight vectors are noted as follows;

- $r_1$: Refusal Vector of the Original Model
- $r_2$: Refusal Vector of the Fine-Tuned Model.
- $W_{l_2}^f$: The projection weight matrices (both the Attention and MLP projection) in the *fine-tuned model* at layer $l_2$ where $l_2$ is the layer index corresponding to the refusal direction (i.e., $r_2$) in the fine-tune model.
- $W_{l_1}$: The projection weight matrices (both the Attention and MLP projection) in the *fine-tuned model* at layer $l_1$ where $l_1$ is the layer index corresponding to the original model's refusal direction (i.e., $r_1$).

Table 2 shows different ways to inject refusal direction in the fine-tuned model. For example, variant A is corresponding to a method under which we inject refusal direction $r_1$ calculated based on the original model into the weight matrix of $W_{l_2}^f$ in the fine-tuned model.

| Variant | Vector Under Consideration | Layer Under Consideration |
|:-------:|:--------------------------:|:-------------------------:|
| A | $r_1$ | $W_{l_2}^f$ |
| B | $r_1$ | $W_{l_1}$ |
| C | $r_2$ | $W_{l_2}^f$ |
| D | $r_2$ | $W_{l_1}$ |
| E | $r_2 - r_1$ | $W_{l_2}^f$ |
| F | $r_2 - r_1$ | $W_{l_1}$ |

Table 2: Variants of weight-based refusal injection considered.

The model we consider for this experiment is the `Llama2-chat-7b` fine-tuned on medical data. We also utilize our general safety metrics along with the GCG jailbreak (Zou et al., 2023b) to evaluate which version of weight steer is the most useful; we record our findings in Table 3.

From Table 3, we can conclude that variant A has the best results to improve the safety of the language models. We utilize the results of variant A and perform all further evaluations on models that had steering performed via Variant A.

| Variant | $\text{ASR}^I_{\text{Adv-Decoding}}$ | $\text{ASR}_{\text{Vanilla}}$ | $\text{ASR}^{\times}_{\text{Adv-Decoding}}$ | GCG |
|---------|------|------|------|-----|
| Base | **1.4** | **0** | **18.4** | **31** |
| A | **2.8** | **0** | **10.8** | **4** |
| B | **3.2** | **1** | **11.6** | **20** |
| C | **2** | **0** | **16.4** | **25** |
| D | **1.8** | **1** | **18.6** | **27** |
| E | **8** | **3** | **13.6** | **4** |
| F | **4.6** | **1** | **14.4** | **18** |

Table 3: Attack Scores (**ASR**) scores of the evaluations. We set the $\alpha = 0.015$ for attention steering $\alpha = 0.01$ for MLP steering for all cases.

## 5.2 SAFETY EVALUATIONS

**General Safety:**  We now evaluate the general safety of the models which were steered via SPEC-TRA and compare the results vis-a-vis the base fine-tuned models. We report the ASR[3] scores (in %) of our safety evaluation in Table 4. From Table 4 we can see that SPECTRA vastly increases the

| Model | Domain | Method | GCG | AdvDecoding | Vanilla | GPTFuzz | TAP | In-Domain |
|-------|--------|--------|-----|-------------|---------|---------|-----|-----------|
| Llama2 | Medical | BASE | 31 | 18.4 | 0 | 8.8 | 22.0 | 28.78 |
| | | SPECTRA | **6** | **9.6** | 0 | **0.0** | **0.0** | **11.56** |
| | Law | BASE | 66 | 29.6 | 0 | 15.0 | 58.0 | 17.0 |
| | | SPECTRA | **43** | **23.2** | 0 | **13.0** | **0.0** | **6.5** |
| | Finance | BASE | 62.0 | 29.0 | 0 | 7.8 | 28.0 | 8.45 |
| | | SPECTRA | **47.0** | 29.0 | 0 | 7.8 | **0.0** | **7.05** |
| Gemma2 | Medical | BASE | 36 | 9.4 | 1 | 68.6 | 36.0 | 20 |
| | | SPECTRA | **0** | **5.6** | **0** | **7.0** | **0.0** | **5.44** |
| | Finance | BASE | 66.0 | 49.2 | 21.0 | 64.7 | 8.0 | 71.83 |
| | | SPECTRA | **1.0** | **16.0** | **0** | **0.0** | 10.0 | **7.04** |

Table 4: Comparison of Base fine-tuned models and SPECTRA across different models and domains. Lower scores indicate better robustness, and the best (lowest) results within each domain block are in bold.

general safety of the language models across the board while being lightweight and simple. Specifically, in the case of both of the variants of the `Gemma2-9b-it`, we see the most drastic increase in the safety capabilities of the language models.

Notably, each model and its respective fine-tuned variants have unique $\alpha$ values and rank of the low rank projections, and mild hyperparameter optimization was performed to achieve the results, see A.1 and A.2 for parameter choices and see A.8 for the layers on which steering has been performed..

**Robustness Against Jailbreaks:**  From Table 4, we see that each model variant's robustness against jailbreaks has increased dramatically. This robustness against jailbreaks is also seen to be method agnostic, safeguarding the model against attacks that need weights (GCG) (Zou et al., 2023b) and attacks that do not need access to model weights (GPTFuzz (Yu et al., 2023), TAP (Mehrotra et al., 2024)).

---

[3]for LLAMA2-7B-CHAT we report the results of $\text{ASR}^{\times}_{\text{Adv-Decoding}}$ due to the high safety risks associated with the method.

**In-Domain Safety:** We also note that SPECTRA performs incredibly well in regards to improving the in-domain safety of language models, without requiring domain-specific data in its procedure. As it stands, SPECTRA is the only method to improve the domain-specific safety of language models without any backpropagation.

## 5.3 COHERENCE EVALUATIONS

**General Coherence:** We test general coherence via conducting zero-shot performance evaluations on HellaSwag, WinoGrande, ARC Challenge, and BoolQ, see Table 5. We note that the impact of SPECTRA on the general coherence of a language model is statistically insignificant, and the method improves safety without much change to a model's general coherence. Notably, we do find that in some cases our method improves a capability; however, this increase is quite insignificant, but could potentially lead to fruitful future work that utilizes our method to possibly increase both the safety and coherence of a model via weight steering. Notably, we do find that `Gemma2-9b-it Finance` sees the most decrease in general coherence vis-a-vis other models tested. This decrease, though quite mild, is accompanied by the largest increase in the model's safety capabilities of all the models tested.

| Model | Domain | RTE | ARC | BoolQ | Winogrande | HellaSwag |
|---|---|---|---|---|---|---|
| Llama2 | **Medical** | 68.5 / 67.5(+1.0) | 45.0 / 45.0(+0.0) | 78.0 / 78.5(−0.5) | 69.0 / 68.5(+0.5) | 60.0 / 58.5(+1.5) |
| | **Law** | 69.5 / 69.0(+0.5) | 44.5 / 44.0(+0.5) | 78.0 / 80.0(−2.0) | 65.5 / 68.0(−2.5) | 57.0 / 57.5(−0.5) |
| | **Finance** | 70.0 / 70.5(−0.5) | 41.5 / 40.5(+1.0) | 81.0 / 79.5(+1.5) | 73.5 / 70.0(+3.5) | 52.0 / 52.0(+0.0) |
| Gemma2 | **Medical** | 72.5 / 72.0(+0.5) | 55.5 / 57.5(−2.0) | 88.0 / 88.0(+0.0) | 72.5 / 71.0(+1.5) | 52.5 / 53.0(−0.5) |
| | **Finance** | 77.0 / 80.5(−3.5) | 53.0 / 52.5(+0.5) | 86.0 / 91.0(−5.0) | 70.5 / 72.5(−2.0) | 53.5 / 53.5(+0.0) |

Table 5: Performance of SPECTRA across different models and domains on standard benchmarks (RTE, ARC, BoolQ, Winogrande, HellaSwag).

**In Domain Coherence:** To further evaluate the effect of SPECTRA, we analyze the domain specific knowledge retention of models that underwent SPECTRA and report our findings in Table 6. We find that models that underwent SPECTRA, retain their domain specific knowledge and do undergo a statistically significant decrease in domain knowledge. This further fortifies the idea that models that undergo SPECTRA denote **no significant change in both general and domain specific coherence**.

| Model | Domain | Medical | | Law | | Finance |
|---|---|---|---|---|---|---|
| | | **PubMedQA** | **MedMCQA** | **Abercrombie** | **Hearsay** | **FinanceBench** |
| Llama2 | Medical | 77.5 / 77.5(+0.0) | 41.7 / 41.7(+0.0) | – | – | – |
| | Law | – | – | 29.4 / 28.4(+1.0) | 63.04 / 65.2(−2.16) | – |
| | Finance | – | – | – | – | 70.0 / 66.67(+3.33) |
| Gemma2 | Medical | 95.5 / 95.5(+0.0) | 52.5 / 52.5(+0.0) | – | – | – |
| | Finance | – | – | – | – | 66.7 / 68.0(−1.3) |

Table 6: Performance of SPECTRA across models and domains. Medical domains are evaluated on PubMedQA and MedMCQA, Law on Abercrombie and Hearsay from LegalBench, and Finance on FinanceBench.

## 5.4 FALSE REFUSAL

Prior literature has shown that directional refusal vector steering has shown to produce refusal on unrelated and safe prompts (O'Brien et al., 2025; Lee et al., 2025). To see whether this finding applies to SPECTRA, we analyze false refusal rates in models that underwent SPECTRA and report our findings, see Table 10.

We compare how different models behave after SPECTRA and analyze whether refusal is disproportionately triggered on benign prompts. To study this, we prompt the models with 100 prompts from the ALPACA (Taori et al., 2023) dataset and calculate the percentage of false refusals before and after SPECTRA. To calculate false refusal rates, we prompt Gemini 2.5 Pro (Comanici et al., 2025) to evaluate model responses and report our findings in Table 10. Overall, we find that SPECTRA produces almost no change to the false refusal rates, except in the case of Gemma2-9b-it Finance, which sees an extremely significant increase in all safety dimensions considered.

| Model | Domain | False Refusal ($\Delta\%$) |
|---|---|---|
| Llama2 | Medical | 0% |
| Llama2 | Law | $-1\%$ |
| Llama2 | Finance | 0% |
| Gemma2 | Finance | 2% |
| Gemma2 | Medical | 0% |

Table 7: Change in False refusal rates after SPECTRA (Post SPECTRA - Pre SPECTRA).

Notable, we record that SPECTRA **does not significantly increase the false refusal rates** in models, unlike prior activation based steering methods (O'Brien et al., 2025).

## 6 RELATED WORK

**Improving Safety of Fine-Tuned Models:** Works have aimed at improving the safety of fine-tuned language models via further preference optimization (Han et al., 2024), curating specific safety data to improve overall safety evaluation by further tuning (Jan et al., 2024), introducing a novel finetuning objective to mitigate fine-tuning-based safety attacks (Qi et al., 2024a), creating safety prefix prompts as backdoors during the fine-tuning procedure (Wang et al., 2024a) or via separating states during the fine-tuning procedure and optimization state drift to prevent alignment degradation (Huang et al., 2024a). All of the aforementioned methods rely on further gradient calculations/tuning or altering the fine-tuning objective or require further domain-specific calibration data , which can pose constraints on the efficacy and cost of the fine-tuning procedure. Our method distinguishes itself by directly injecting a steering vector into a low-rank safety-related space of the fine-tuned variant, making it simple, efficient, and having low spillover onto other capabilities.

**Refusal Steering Vectors:** Prior work has identified the refusal steering vector (Arditi & Obeso, 2023) and utilized its significance to create a white box jailbreak method that incurs low cost and enjoys a high Attack Success Rate. Other works have discovered such steering vectors via the use of sparse autoencoders (O'Brien et al., 2024). Prior research has also utilized steering vectors to mitigate false refusals in language models (Wang et al., 2024b). While some works have applied steering vectors to steer refusal and improve model safety (O'Brien et al., 2025), they have shown to have significant downsides on the general coherence of the language model and/or require inference time interventions (Lee et al., 2025) to steer model activations.

## 7 DISCUSSION

In this work, we introduced a novel method, SPECTRA, for eliciting safety behaviors in fine-tuned language models that suffer from a degradation of safety even after fine-tuning on benign data. From our experimentation and evaluations, we find that SPECTRA leads to no significant downsides while aiding the safety of fine-tuning language and providing robustness against various jailbreaks. We find that this finding holds true for a multitude of model fine-tunes and domains. Furthermore, our method aids the model's domain-specific safety and doesn't degrade the domain-specific coherence of the language model. Our method, while being simple, remains very efficient, requiring only changes to at most two projection matrices in the language model. Fundamentally, the method aims to steer the weights of a language model to aid refusal; however, as prior work (Wei et al., 2024) has noted, a significant portion of the ranks in the language model weights do not contribute to the safety of the language. As our method relies on orthogonalizing the refusal vector with respect to the weights, to mitigate potential downsides, we calculate a low-rank projection of the weight matrices such that the safety-related ranks are preserved in the low-rank approximation. Intuitively, this could mitigate the potential downsides of the weight steer as we see various evaluations.

**Future Work:** We believe that our method could be applied to diverse applications beyond fine-tuning to steering vectors in general and could lead to exciting future work. Furthermore, we believe that our method can see immediate application in deployed language models due to its simplicity and cost-effectiveness, which could red team the method against a variety of jailbreaks and could elucidate the method's impact on a variety of applications/capabilities.

**Limitations :** We do acknowledge that due to the ever-evolving landscape of jailbreaks, a variety of such adversarial attacks could be formulated to mitigate the robustness and guardrails that our method provides. Future work in generalizing our method is needed to ensure safer models. Furthermore, as there are many capabilities in modern large language models that are of interest, the potential downsides of our method need to be further evaluated on such capabilities to provide an expansive view of the impact of weight steering, although our work does insinuate that overall coherence of the language model is not impacted, many nuanced capabilities not studied in this work have the potential to be impacted and such evaluations are necessary to generalize our method.

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

# A APPENDIX

## A.1 $\alpha$ VALUES OF THE REPORTED EXPERIMENTS

The $\alpha$ values used in the paper are model-dependent and were discovered via mild hyperparameter tuning. The following values of $\alpha$ were used for each model variant:

| Model family | Domain | Attention$_\alpha$ | MLP$_\alpha$ |
|---|---|---|---|
| LLAMA-2 CHAT | Medical | 0.07 | 0.05 |
| LLAMA-2 CHAT | Law | 0.1 | 0.03 |
| LLAMA-2 CHAT | Finance | 0.1 | 0.05 |
| GEMMA2-9B-IT | Medical | 0.01 | 0.00 |
| GEMMA2-9B-IT | Finance | 0.01 | 0.00 |

**Hyperparameter sensitivity** : We do a sweep search and report the ASR scores of the GCG (Zou et al., 2023b) attack on the model vs $\alpha$ for the medical fine-tune of llama-2-chat (Rohanian et al., 2024). We report two cases: In the first case, we keep $\alpha$ values for the MLP constant at $\alpha = 0.05$ and sweep the $\alpha$ values for the attention heads, see 1. Similarly we repeat for $\alpha$ of attentions heads = 0.07 and report the sweep for $\alpha$ values for MLPs, see 2.

## A.2 RANK OF THE LOW MATRIX FOR WEIGHT STEERING

We now report the rank of the matrices used for the weight steer. They are as follows:

| Model family | Domain | Attention | MLP |
|---|---|---|---|
| LLAMA-2 CHAT | Medical | 2096/4096 | 2096/4096 |
| LLAMA-2 CHAT | Law | 3096/4096 | 3096/4096 |
| LLAMA-2 CHAT | Finance | 2096/4096 | 2096/4096 |
| GEMMA2-9B-IT | Medical | 1048/2048 | 2048/3048 |
| GEMMA2-9B-IT | Finance | 1048/2048 | 2048/3048 |

Table 8: Reporting of the low rank matrix. Reported as new rank/ original rank.

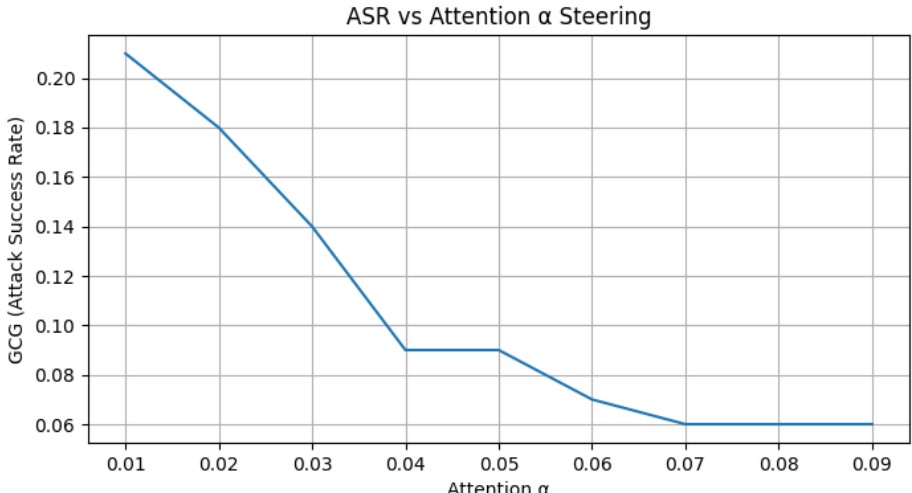

Figure 1: ASR of GCG attacks vs $\alpha$ for attention heads, with fixed $\alpha$ for MLPs.

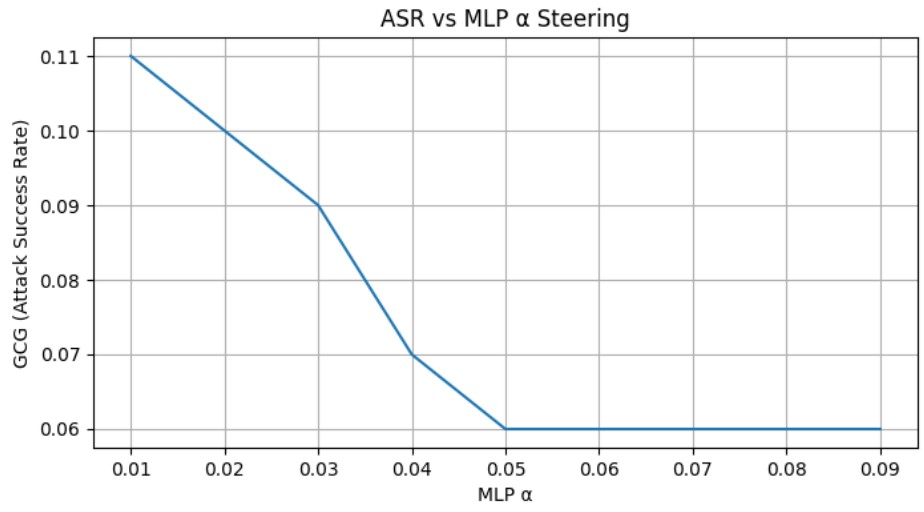

Figure 2: ASR of GCG attacks vs $\alpha$ for mlp, with fixed $\alpha$ for attention.

### A.3 REFUSAL DIRECTION SELECTING ALGORITHM

We borrow the refusal direction selection algorithm from Arditi et al. (2024). Given a collection of difference-in-means vectors, denoted as $\{\mathbf{r}_i^{(l)} | i \in I, l \in [L]\}$, we evaluate the following key metrics:

- **bypass_score**: Measures the average refusal rate on the validation set of harmful prompts ($\mathcal{D}_{\text{harmful}}^{(\text{val})}$) when applying directional ablation to $\mathbf{r}_i^{(l)}$.

- **induce_score**: Assesses the average refusal rate on the validation set of harmless prompts ($\mathcal{D}_{\text{harmless}}^{(\text{val})}$) when the activation addition of $\mathbf{r}_i^{(l)}$ is applied.

- **kl_score**: Computes the average Kullback-Leibler (KL) divergence between the model's probability distributions at the final token position when evaluated on $\mathcal{D}_{\text{harmless}}^{(\text{val})}$ with and without directional ablation of $\mathbf{r}_i^{(l)}$.

To identify the optimal direction $\mathbf{r}_{i*}^{(l^*)}$, we select the vector with the lowest `bypass_score`, while ensuring the following constraints are met:

- `induce_score` $> 0$
  - Ensures that the selected direction is capable of inducing a refusal response.
- `kl_score` $< 0.1$
  - Prevents the selection of directions that excessively alter model behavior on benign prompts.
- $l < 0.8L$
  - Restricts the selection to earlier layers, avoiding interference with unembedding representations.

### A.4 GCG ATTACK DETAILS

We borrow and modify the methodology of Wei et al. (2024) to generate adversarial suffixes, which is: Run the GCG attack Zou et al. (2023b) for $500$ iterations, with adversarial string initiated as "! ! ! ! ! ! ! ! ! ! ! ! ! ! ! ! ! ! ! ! !" and a batch size of $256$, top-$k$ as $128$, with optimization over the fine-tuned models, with the system prompts removed, for three independent trials. We then identify the top three suffixes with the highest attack success rates on AdvBench, and use them in our evaluation.

### A.5 DETAILS OF ZERO-SHOT EVALUATIONS

1. **ARC-Challenge:**
   (a) **Downstream Task:** Science Question Answering.
   (b) **Overview:** This metric gauges model performance on the ARC-Challenge portion of the AI2 Reasoning Challenge dataset. It comprises grade-school science questions that necessitate complex reasoning and an in-depth understanding of scientific principles[4].

2. **HellaSWAG:**
   (a) **Downstream Task:** Commonsense Reasoning.
   (b) **Overview:** HellaSWAG is designed to test commonsense reasoning capabilities. It presents a context followed by several multiple-choice endings, with the objective of selecting the most plausible continuation. The dataset challenges models to interpret and reason about everyday situations[5].

3. **WinoGrande:**
   (a) **Downstream Task:** Commonsense Reasoning.
   (b) **Overview:** WinoGrande is a large-scale dataset for assessing commonsense reasoning. Presented as a fill-in-the-blank task with binary choices, the aim is to select the appropriate option, demanding robust commonsense understanding while mitigating dataset-specific biases[6].

4. **BoolQ:**
   (a) **Downstream Task:** Yes/No Question Answering.
   (b) **Overview:** BoolQ is a dataset focused on yes/no questions, featuring 15,942 naturally occurring examples. Each instance comprises a question, a passage, and the corresponding answer, with optional contextual information such as the page title. The setup is akin to text-pair classification tasks found in natural language inference research[7].

---

[4] Further details can be found at `https://allenai.org/data/arc`.

[5] Additional information is available at `https://huggingface.co/datasets/Rowan/hellaswag`.

[6] Further information is available at `https://huggingface.co/datasets/winogrande`.

[7] More details can be found at `https://github.com/google-research-datasets/boolean-questions`.

5. **RTE (Recognizing Textual Entailment):**

   (a) **Downstream Task:** Textual Entailment.

   (b) **Overview:** The RTE task involves deciding whether a hypothesis can be logically inferred from a given premise. The dataset consists of sentence pairs, where the goal is to classify each pair as either "entailment" (if the hypothesis logically follows from the premise) or "not entailment" (if it does not)[8].

## A.6 IMPACT OF SPECTRA ON MULTI-TURN DIALOGUE

We now measure the change in the multi-turn dialogue capabilities in models that underwent SPEC-TRA. To measure this we test the capabilities of fine-tuned models vs models that underwent SPEC-TRA on MultiChallenge (Sirdeshmukh et al., 2025) and report our findings in 9.

Table 9: Model Performances on MultiChallenge

| Model | Method | Inference Memory | Self Coherence | Instruction Retention | Reliable Version Editing |
|---|---|---|---|---|---|
| *Finance Domain* | | | | | |
| Gemma2 | Base | 2.15 | 9.09 | 29.41 | 0.00 |
| Gemma2 | SPECTRA | 4.42 | 10.00 | 13.04 | 5.00 |
| Llama2 | Base | 4.42 | 2.04 | 14.49 | 14.63 |
| Llama2 | SPECTRA | 5.31 | 10.00 | 5.80 | 9.76 |
| *Medical Domain* | | | | | |
| Gemma2 | Base | 8.85 | 6.00 | 10.14 | 4.88 |
| Gemma2 | SPECTRA | 4.42 | 14.00 | 23.53 | 4.88 |
| Llama2 | Base | 3.54 | 10.00 | 17.39 | 9.76 |
| Llama2 | SPECTRA | 5.31 | 6.12 | 20.59 | 0.00 |
| *Law Domain* | | | | | |
| Llama2 | Base | 7.96 | 18.00 | 20.29 | 7.32 |
| Llama2 | SPECTRA | 5.31 | 6.00 | 14.49 | 7.32 |

## A.7 COMPARISON TO OTHER STEERING

We now comparing SPECTRA to activation steering (activation addition) (Arditi et al., 2024) and simple weight steering seen in section 5.1. We report our findings in 10.

## A.8 REFUSAL DIRECTIONAL CHANGE AFTER FINE-TUNING

We report that the fine-tuned models notice a change in source position of their refusal directions. These changes are noted as follows:

Note, we refer to $l_2$ and $l_1$ as the layer of the refusal direction in the fine-tuned and original model, respectively, $tp_2$ and $tp_1$ are the token positions of each refusal vector, respectively, as well.

---

[8]Additional details are available at `https://huggingface.co/datasets/nyu-mll/glue#rte`.

Table 10: Change in False Rates (After Steering - Before) for simple steering and activation addition.

| Model | Domain | Method | False Refusal ($\Delta$%) |
|---|---|---|---|
| **Gemma2** | Finance | Simple Weight Steering | 91% |
| | | Activation Addition | 7% |
| | Medical | Simple Weight Steering | 88% |
| | | Activation Addition | 5% |
| **Llama2** | Finance | Simple Weight Steering | 93% |
| | | Activation Addition | 9% |
| | Law | Simple Weight Steering | 97% |
| | | Activation Addition | 6% |
| | Medical | Simple Weight Steering | 98% |
| | | Activation Addition | 8% |

| Model | Domain | $l_2/l_1$ | $tp_2/tp_1$ |
|---|---|---|---|
| Llama2 | Medical | 13/14 | $-5/-1$ |
| | Law | 12/14 | $-2/-1$ |
| | Finance | 12/14 | $-5/-1$ |
| Gemma2 | Medical | 21/31 | $-1/-1$ |
| | Finance | 31/31 | $-5/-1$ |

Table 11: Changes in Source Position of the Refusal Direction vectors

