# OpenReview forum: "Fixing What Fine-Tuning Breaks: A Simple and Efficient Method to Improve Safety Post Domain Adaptation"
_ICLR.cc/2026/Conference — Submitted to ICLR 2026_

### Official Review · Reviewer_e64L · 2025-11-01

**Soundness:** 1
**Presentation:** 2
**Contribution:** 1
**Rating:** 2
**Confidence:** 5

**Summary:**

The paper addresses safety degradation in LLMs after domain-specific fine-tuning. It proposes SPECTRA, a gradient-free method that restores safety by identifying a refusal direction from activation differences between harmful and benign prompts, then injecting it into a safety-critical subspace via activation-aware SVD. Tests on Llama2-7B and Gemma2-9B across multiple domains show improved safety and jailbreak resistance with minimal impact on performance.

**Strengths:**

1. The paper considers the critical and timely problem of efficient safety alignment for LLMs.
2. The empirical results demonstrate the method's positive impact when compared against the base model.
3. The exploration of safety in specialized domains like finance and law is a valuable extension.

**Weaknesses:**

1. The experimental evaluation is limited, as it only compares the proposed method against the base model. To properly assess its contribution, a comparison with established safety alignment baselines is necessary.
2. The contribution appears to be an incremental extension of existing techniques (e.g., Wei et al. (2024), Chhabra & Khalili (2025)). The paper should more clearly show the distinctions from prior work and provide deeper insights.
3. The academic writing needs revision for clarity and precision. Several key claims are not sufficiently supported by the empirical results.

**Questions:**

You claim that the impact on general coherence is "statistically insignificant." To substantiate this claim, could you please provide the specific statistical test results (e.g., p-values) from your benchmark evaluations that support this conclusion?

---

### Official Review · Reviewer_zaiV · 2025-11-01

**Soundness:** 2
**Presentation:** 1
**Contribution:** 1
**Rating:** 2
**Confidence:** 3

**Summary:**

The paper examines how an LLM’s safety alignment often degrades after fine-tuning, even when the fine-tuning data are harmless. It introduces SPECTRA, which computes a safety-critical low-rank direction and injects it into the model’s weights, steering the model toward safer outputs while largely preserving capabilities on domain-specific tasks.

**Strengths:**

- This paper addresses a real and urgent problem in LLMs: fine-tuning can break safety, which is a major deployment concern.
- It offers a practical solution: no retraining and no domain-specific safety data are required.

**Weaknesses:**

1. This paper requires thorough proofreading, as the current writing makes the presentation of this work incomplete.
- Abstract does not end with a period.
- difference-in-means belrose2023diffinmeans …
- both the the Attention and MLP
- safty → safety
- neccessary → necessary
- Harmbech → Harmbench
- coherance → coherence
- The paper includes several tables (1-4) that present information better suited for inline text or an appendix.


2. You must rephrase all text in your own words, whether it presents fundamental facts or cites a source.
- Section 2 should be rewritten.
- Portions of your text appear to be copied directly from [1]. It also appears that [1] copied much of its background from [2], which must not happen.


3. The most critical problem with this paper is that its contribution appears insignificant.
This paper appears to combine existing ideas from [2] and [3]. If that is not the case, please correct me if I’m mistaken.
- The experiments present results for SPECTRA, but without baseline comparisons, it is unclear whether the proposed idea is actually effective.
    - I believe it is important to include a comparison with [2].
- “SPECTRA is the only method to improve the domain-specific safety of language models without any backpropagation.”
    - This claim seems overstated unless you demonstrate that other general, non-backpropagation methods (e.g., [2], …) are less effective in the same domain-specific setting.


---


[1] Chhabra, Vishnu Kabir, and Mohammad Mahdi Khalili. "Towards Understanding and Improving Refusal in Compressed Models via Mechanistic Interpretability." *arXiv preprint arXiv:2504.04215* (2025).

[2] Arditi, Andy, et al. "Refusal in language models is mediated by a single direction." Advances in Neural Information Processing Systems 37 (2024): 136037-136083.

[3] Wei, Boyi, et al. "Assessing the brittleness of safety alignment via pruning and low-rank modifications." arXiv preprint arXiv:2402.05162 (2024).

**Questions:**

- [Table 6] Did you perform any additional analysis explaining why Variant A performs best?
- [Table 6] Does “Base” refer to the fine-tuned model or the original LLM?

---

> ### Author Response · Authors · 2025-11-20
> **Response to review by Reviewer zaiV**
>
> We thank the reviewer for this detailed and caring review. We sincerely appreciate that the reviewer found our method proposed to be a practical approach to aiding modern AI safety.
>
> >“This paper requires thorough proofreading, as the current writing makes the presentation of this work incomplete”
>
> We have adjusted typos highlighted and fixed some grammatical/structural issues with the paper, We have also adjusted the tables 1-4 to be included in the main paper as inline text.
>
> >“You must rephrase all text in your own words, whether it presents fundamental facts or cites a source”
>
> We have made the changes to Section 2 of the paper to highlight this concern.
>
> >“The most critical problem with this paper is that its contribution appears insignificant. This paper appears to combine existing ideas from [2] and [3]. If that is not the case, please correct me if I’m mistaken”
>
> Thank you for pointing this out, although the works [2] and [3] provide inspiration for the methodology proposed in this paper. Our work is significantly different in terms of contribution and methodology. More specifically:
>
> [2] provides a key interpretability finding: the refusal feature in language models is encapsulated via a single  direction in the activation space. This work however does not propose a methodology to improve the safety of large language models and rather just presents a finding about the refusal feature in language models.
>
> [3] provides an insight into the sparsity of the safety related ranks/neuron in modern large language models. They do so by utilizing compression such as pruning and low rank approximation to isolate key ranks/neurons. The methodology used for isolation key ranks in the paper: ActSVD is not a novel methodology introduced in the work and has been a long standing methodology utilized in DNN compression[4,5,6,7]. We just utilized a low rank based approach to minimize the ranks in the column space of W which have non-zero dot products with refusal direction, i.e, we aim to minimize the ranks of W which have utility/non-safety related functionality. To minimize the spillover onto other functionality when steering refusal as seen in [8] and [9].
>
> We believe the novelty of the approach we propose is in not only the weight steering methodology proposed but also the fact that such a weight steering methodology has minimum impact on false positives compared to the other baselines.
> >“
> The experiments present results for SPECTRA, but without baseline comparisons, it is unclear whether the proposed idea is actually effective.
>  believe it is important to include a comparison with [2].
> "SPECTRA is the only method to improve the domain-specific safety of language models without any backpropagation.”
> This claim seems overstated unless you demonstrate that other general, non-backpropagation methods (e.g., [2], …) are less >effective in the same domain-specific setting”
>
> Thank you for pointing this out. To address this concern we compared out methodology with [1] and [2] in terms of false refusal rates/over refusal. For [2] we took the standard activation steering approach denoted in “Activation Addition” section of [2]. The results are as follows:
>
>
> Baseline Activation Addition[2], false refusal rates changes (post activation addition - prior) on Alpaca:
> | Model | Medical | Finance | Law |
> | :--- | :---: | :---: | :---: |
> | **Llama 2** | 98% | 93% | 97% |
> | **Gemma 2** | 88% | 91% | - |
>
> We also compared the false positive results for AIRD[1] although this method is originally meant for pruned models( section 4 and section 6 of [1]). As for fine-tuned models, the weight matrices have full rank hence the possibility of perturbing non-safety related ranks increases and hence we see a degradation in performance(increase in false positives) with this method. As SPECTRA, maps the weight matrix to a low rank approximation with safety related ranks, the spillover to non-safety related ranks is minimized and hence we see minimum downsides with SPECTRA.
>
> AIRD[1], false refusal rates:
>
> | Model | Medical | Finance | Law |
> | :--- | :---: | :---: | :---: |
> | **Llama 2** | 8% | 9% | 6% |
> | **Gemma 2** | 5% | 7% | - |
>
> We plan to add these experiments to the main paper based on the feedback from the reviewer.
>
> Again, we would like to thank the reviewer for their review and would ask the following question: Do these added experiments and changes highlight the contribution further and address your concerns? If not, is there anything else you would like us to consider in the draft?

---

> > ### Author Response · Authors · 2025-11-20
> > **Response to review by Reviewer zaiV (2/2)**
> >
> > [1] Chhabra, Vishnu Kabir, and Mohammad Mahdi Khalili. "Towards Understanding and Improving Refusal in Compressed Models via Mechanistic Interpretability." arXiv preprint arXiv:2504.04215 (2025).
> >
> > [2] Arditi, Andy, et al. "Refusal in language models is mediated by a single direction." Advances in Neural Information Processing Systems 37 (2024): 136037-136083.
> >
> > [3] Wei, Boyi, et al. "Assessing the brittleness of safety alignment via pruning and low-rank modifications." arXiv preprint arXiv:2402.05162 (2024).
> >
> > [4]Yuan, Zhihang, et al. "Asvd: Activation-aware singular value decomposition for compressing large language models." arXiv preprint arXiv:2312.05821 (2023).
> >
> > [5]Emily L Denton, Wojciech Zaremba, Joan Bruna, Yann LeCun, and Rob Fergus. Exploiting linear structure within convolutional networks for efficient evaluation. Advances in neural information processing systems, 27, 2014
> >
> > [6]Vadim Lebedev, Yaroslav Ganin, Maksim Rakhuba, Ivan Oseledets, and Victor Lempitsky. Speeding-up convolutional neural networks using fine-tuned cp-decomposition.2014
> >
> > [7]Marcin Moczulski, Misha Denil, Jeremy Appleyard, and Nando de Freitas. Acdc: A structured efficient linear layer., 2015
> > [8] Kyle O’Brien, David Majercak, Xavier Fernandes, Richard Edgar, Blake Bullwinkel, Jingya Chen,
> > Harsha Nori, Dean Carignan, Eric Horvitz, and Forough Poursabzi-Sangdeh. Steering language
> > model refusal with sparse autoencoders, 2025
> > [9] Bruce W. Lee, Inkit Padhi, Karthikeyan Natesan Ramamurthy, Erik Miehling, Pierre Dognin, Manish Nagireddy, and Amit Dhurandhar. Programming refusal with conditional activation steering,
> > 2025

---

### Official Review · Reviewer_hfRs · 2025-11-02

**Soundness:** 3
**Presentation:** 3
**Contribution:** 3
**Rating:** 6
**Confidence:** 2

**Summary:**

This paper discusses the issue that safety-aligned language models often lose their safety properties after being fine-tuned on domain-specific data, even when the data itself is harmless. The authors introduce SPECTRA, a lightweight method that restores safety after fine-tuning by injecting a refusal steering vector into a low-rank portion of the model’s weights. Unlike methods such as preference optimization or RLHF, SPECTRA does not require additional training, domain-specific safety data, or large computational resources. The experiments, conducted on medical, legal, and financial versions of LLaMA and Gemma, show that the method substantially lowers attack success rates in both standard and jailbreak evaluations, maintains task performance in general and domain settings, and shows almost no increase in false refusals.

**Strengths:**

* The paper proposes a method that improves safety without any additional training by modifying the model weights directly.
* It demonstrates the effectiveness of the method through experiments across multiple domains.
* It also reports results on general language capability, showing the negative results as well, rather than only highlighting the improvements.

**Weaknesses:**

* Limitations of refusal-vector–based safety
- Since the method strengthens the “refusal direction,” it mainly makes the model better at rejecting harmful queries, but it does not actually mean the model understands or reasons about harmful content. It would be helpful to include experiments that evaluate this limitation more directly.
* Insufficient evaluation of other capabilities
- The paper reports coherence and general task performance, but it would be more convincing if it also evaluated qualitative abilities such as reasoning, multi-turn dialogue quality, and instruction-following faithfulness.
* Hyperparameter choices are mostly empirical
- The selection of alpha and the rank values seems empirical. Since the cost of SVD increases as the steering scale becomes larger, it would be useful to include an analysis of the runtime or compute impact when applying this method to larger models.

**Questions:**

NA

---

> ### Author Response · Authors · 2025-11-20
> **Response to review by Reviewer hfRs**
>
> Firstly, we would like to thank the reviewer for their detailed and caring review. We sincerely appreciate that you found our work meaningful and transparent
>
> >Limitations of refusal-vector–based safety
>
> Can you please elaborate further on this? We highlighted the limitations of such a method in the discussion sections and would like understand how we can be further transparent on this concern.
>
> >Since the method strengthens the “refusal direction,” it mainly makes the model better at rejecting harmful queries, but it does not actually mean the model understands or reasons about harmful content. It would be helpful to include experiments that evaluate this limitation more directly. Insufficient evaluation of other capabilities
> The paper reports coherence and general task performance, but it would be more convincing if it also evaluated qualitative abilities such as reasoning, multi-turn dialogue quality, and instruction-following faithfulness.
>
> To address these concern, we perform an experiment on analyzing the change in multiturn capability of language models going through the changed methodology. We benchmark this on MultiChallenge [1] benchmark and note the following results.
>
> | Model | Method | Inference Memory | Self Coherence | Instruction Retention | Reliable Version Editing |
> | :--- | :--- | :---: | :---: | :---: | :---: |
> | Gemma2 Finance | Base | 2.15 | 9.09 | 29.41 | 0.00 |
> | Gemma2 Finance | SVDAIRD | 4.42 | 10.00 | 13.04 | 5.00 |
> | Gemma2 Med | Base | 8.85 | 6.00 | 10.14 | 4.88 |
> | Gemma2 Med | SVDAIRD | 4.42 | 14.00 | 23.53 | 4.88 |
> | Llama2 Finance | Base | 4.42 | 2.04 | 14.49 | 14.63 |
> | Llama2 Finance | SVDAIRD | 5.31 | 10.00 | 5.80 | 9.76 |
> | Llama2 Law | Base | 7.96 | 18.00 | 20.29 | 7.32 |
> | Llama2 Law | SVDAIRD | 5.31 | 6.00 | 14.49 | 7.32 |
> | Llama2 Med | Base | 3.54 | 10.00 | 17.39 | 9.76 |
> | Llama2 Med | SVDAIRD | 5.31 | 6.12 | 20.59 | 0.00 |
>
> From this we see that for models like Gemma2-9b-it which have been highlighted to have good reseasoning capabilities see that although some multi turn capabilities are reduced, others are increased. In case of llama2-7b-chat this is also seen. However as both the language models considered in the work don’t report high capabilities as seen in [1], we believe future work to further study the efficacy of our method on multi turn ability is needed.
> We plan to add this experiment to the main text of the paper to address this concern.
>
> >Hyperparameter choices are mostly empirical
> The selection of alpha and the rank values seems empirical. Since the cost of SVD increases as the steering scale becomes larger, it would be useful to include an analysis of the runtime or compute impact when applying this method to larger models.
>
> Thank you for pointing this out, we would like to note that this is a natural outcome of injecting a feature vector into the model weight in this case. However in case of the rank, we intentionally highlighted the cases where we kept 25% and 50% of the weight ranks. This is done so to highlight that the method utilized can still provide good results even for many ranks. As we do the low rank approximation considering the safety related data, we aim to eliminate utility critical ranks which are sparse hence allowing for a wide range of rank choices as shown in the work . Furthermore, as highlighted by the reviewer the cost of SVD scales with the model size, however as we are only applying SVD to at most two weight matrices in the language model, we believe this compute tradeoff to be minimal when compared to other backpropagation such as supervised fine-tuning, RLHF, DPO etc.
>
> Again, we would like to thank the reviewer for their detailed reviewed and would like to ask: does the further experimentation address the concern you highlighted and any other clarifications/experimentation you would like us to address?
>
> [1]Sirdeshmukh, Ved, et al. "Multichallenge: A realistic multi-turn conversation evaluation benchmark challenging to frontier llms.
>
> [2] Wei, Boyi, et al. "Assessing the brittleness of safety alignment via pruning and low-rank modifications." arXiv preprint arXiv:2402.05162 (2024).
>
> [3]Yu, Mengxia, et al. "The super weight in large language models

---

### Official Review · Reviewer_5ZCC · 2025-11-06

**Soundness:** 3
**Presentation:** 3
**Contribution:** 3
**Rating:** 6
**Confidence:** 4

**Summary:**

This paper introduces SPECTRA (Scalable Projection-based Elicitation of Coherence and Trustworthy Refusal Alignment), a novel, computationally inexpensive, and efficient weight shifting methodology to counteract the degradation of safety in safety-aligned large language models (LLMs) after domain-specific fine-tuning. The degradation often occurs even when fine-tuning is performed on benign data. Unlike prior realignment methods which require computationally expensive preference optimization or domain-specific alignment data , SPECTRA works by directly injecting a refusal steering vector (calculated via difference-in-means on the base model) into a low-rank, safety-related subspace of the fine-tuned model's weight matrices.
The authors demonstrate that SPECTRA significantly improves general and in-domain safety (measured by Attack Success Rate, ASR) and robustness against jailbreaks (GCG, GPTFuzz, TAP) across Llama2-7B-Chat and Gemma2-9B-IT models fine-tuned for Medical, Law, and Finance domains. Crucially, the method is shown to incur statistically insignificant changes to the model's general and domain-specific coherence and maintains low false refusal rates, addressing common drawbacks of activation-based steering.

**Strengths:**

* Significance & Efficiency: SPECTRA addresses a critical, real-world issue in deploying domain-adapted LLMs (safety degradation) with an exceptionally simple, efficient, and data-light method, requiring no new domain-specific safety data or further gradient calculations. This is a major advantage over gradient-based methods.
* Quality & Robustness: The method significantly improves safety across the board for multiple LLM families (Llama2, Gemma2) and domains (Medical, Law, Finance), providing enhanced robustness against a diverse set of advanced jailbreaking attacks (GCG, GPTFuzz, TAP).
* Clarity & Non-Invasiveness: The theoretical link to activation steering (Theorem 1) provides a clear interpretability anchor. Furthermore, the empirical evidence showing statistically insignificant changes in both general and domain-specific coherence and low false refusal rates successfully mitigates major concerns associated with prior directional steering techniques.
* Originality of Application: The successful application of a base model's refusal vector (r_1) to a fine-tuned model's optimal layer (W_{l2}^f) is a novel finding in the context of weight steering for post-finetuning safety, and Variant A's effectiveness is a key result.

**Weaknesses:**

1. Mechanism Detail and Intuition: The explanation for why Variant A (r_1  injected into W_{l2}^f) is superior to other variants, especially Variant C (r_2 injected into W_{l2}^f ), is currently based solely on empirical results (Table 6). A deeper mechanistic interpretability analysis is needed. The authors should hypothesize why the base model's refusal vector (r_1) retains or even gains efficacy over the fine-tuned model's vector (r_2) in the new weight space, especially in the layer l_2
2. Hyperparameter Sensitivity and Selection: The paper mentions that α values and low-rank p values are model-dependent and found via "mild hyperparameter tuning" (Tables A.1, A.2). This opaque process is a critical weakness. The authors should include a sensitivity analysis (e.g., a simple sweep plot of ASR vs. α and ASR vs. p) to demonstrate that the optimal settings are not highly brittle or require extensive tuning, which would negate the "simple and efficient" claim.
3. Missing Control/Comparison to Full W′ : Equation 5 (W' =W+αrr ^TW) is a special case of SPECTRA when the projection Π is the identity matrix. The main premise of using
$\tilde{W} = \Pi W$ is to mitigate potential downsides on coherence by only steering in the safety-critical subspace Π. The current results (Table 6) use the full W in the steering (Eq. 5) to find the best practice (Variant A) and then implicitly use SPECTRA for all subsequent tables (Tables 7, 8, 9). The paper must explicitly compare the performance (ASR vs. Coherence trade-off) of the full SPECTRA (Eq. 3) versus the simpler steering (Eq. 5) to empirically justify the necessity of the low-rank projection  \tilde{W}

4. Refusal Layer/Token Position Explanation: The layers selected for steering (l_2) are detailed in the appendix (Table 12). The authors should discuss why fine-tuning causes a shift in the refusal layer (e.g., Llama2 Medical: 13/14) and what the mechanistic implication of this shift is.

**Questions:**

1. Justification for  \tilde{W} (The Low-Rank Projection): Table 6, which determines the best practice (Variant A), uses the simpler steering:
$W' = W + \alpha r r^T W$. The core of SPECTRA is $W' = W + \alpha r r^T \tilde{W} $(Equation 3). Can the authors provide a direct comparative evaluation between SPECTRA (Eq. 3) and the simpler steering (Eq. 5), specifically showing the coherence retention benefit that \tilde{W}  is supposed to provide? Without this direct comparison, the necessity of the low-rank projection, which adds complexity (ActSVD computation, rank selection), is not sufficiently justified.

2. Mechanistic Insight into Variant A's Success: Why is Variant A (r_1 into W_{l2}^f , i.e., base model refusal vector in fine-tuned model's optimal layer) consistently the most effective? Specifically, why is it superior to Variant C (r_2 into W_{l2}^f), which uses the refusal direction calculated from the fine-tuned model itself? Does this suggest that the fine-tuning process damages the interpretability/saliency of the refusal direction more than it damages the original weights, making the base model's vector a purer signal? Please provide a detailed hypothesis.

3. Hyperparameter Sensitivity and Generalization: Please provide the sensitivity analysis plots (ASR and coherence metric changes vs. α) for at least two different models/domains (e.g., Llama2 Medical and Gemma2 Finance) to demonstrate that the optimal α and rank p values are stable. How would a practitioner determine optimal α and p without running extensive red teaming evaluations?

4. In-Domain Safety Benchmarks: For the Law and Finance domains, the authors had to create custom safety datasets (FinSafeEval, LawSafeEval). Could the authors briefly describe the nature of the harmful prompts in these datasets? For instance, do the Law prompts relate to illegal advice, and the Finance prompts to fraud/manipulation? This context is crucial for understanding the "In-Domain" safety results.

**Details Of Ethics Concerns:**

The paper involves generating and evaluating responses to harmful instructions (jailbreaking, adversarial attacks) using LLMs to measure safety. While this is standard red-teaming practice, it touches on potential safety risks, warranting a review.

---

> ### Author Response · Authors · 2025-11-24
> **Official Comment by Authors**
>
> Firstly, we would like to thank the reviewer for their careful, detailed and caring review. We sincerely appreciate that you found our work meaningful and impactful.
>
> >Mechanism Detail and Intuition: The explanation for why Variant A (r_1 injected into W_{l2}^f) is superior to other variants, especially Variant C (r_2 injected into W_{l2}^f ), is currently based solely on empirical results (Table 6). A deeper mechanistic interpretability analysis is needed. The authors should hypothesize why the base model's refusal vector (r_1) retains or even gains efficacy over the fine-tuned model's vector (r_2) in the new weight space, especially in the layer l_2
>
> Thank you for pointing this out, we would first like to mention that this is an observation that we have presented in the work. We drew this conclusion from the empirical findings presented in the work. As for why the refusal vector (r_1) of the base model outperforms the fine-tuned models (r_2), we believe this due to the fact the original refusal vector interacts with more features in the language model, this interaction is thus reduced due to domain adaption. Especially in regards to general safety, we believe the fine-tuning objective perturbs the interaction of the refusal vector with harmful attributes hence we find that the original model’s refusal vector performs better than the fine-tuned models. We retracted from including this hypothesis into the work due to a lack of evidence and difficulty of setting up such an experiment, however if the reviewer believes adding this explanation to the work will improve the paper then we will add it.
>
> >“Hyperparameter Sensitivity and Selection: The paper mentions that α values and low-rank p values are model-dependent and found via "mild hyperparameter tuning" (Tables A.1, A.2). This opaque process is a critical weakness. The authors should include a sensitivity analysis (e.g., a simple sweep plot of ASR vs. α and ASR vs. p) to demonstrate that the optimal settings are not highly brittle or require extensive tuning, which would negate the "simple and efficient" claim”
>
> For the ranks presented in the paper, they were chosen arbitrarily, following common low rank choices as seen in [1]. However for alpha, we do agree a ASR vs  α plot would provide more credibility to our claims.
>
> We do a sweep search and report the ASR scores of the GCG attack on the model vs α for the medical fine-tune of llama-2-chat. We report two cases:
>
> | Attention α | GCG (Attack Success Rate) |
> | :---: | :---: |
> | 0.01 | 0.21 |
> | 0.02 | 0.18 |
> | 0.03 | 0.14 |
> | 0.04 | 0.09 |
> | 0.05 | 0.09 |
> | 0.06 | 0.07 |
> | 0.07 | 0.06 |
> | 0.08 | 0.06 |
> | 0.09 | 0.06 |
>
> In the first case, we keep α values for the mlp constant at α = 0.05 and sweep the  α  values for the attention heads
>
> | MLP α | GCG (Attack Success Rate) |
> | :---: | :---: |
> | 0.01 | 0.11 |
> | 0.02 | 0.10 |
> | 0.03 | 0.09 |
> | 0.04 | 0.07 |
> | 0.05 | 0.06 |
> | 0.06 | 0.06 |
> | 0.07 | 0.06 |
> | 0.08 | 0.06 |
> | 0.09 | 0.06 |
>
> In the second case we sweep the α values for the MLP layer and keep the attention head α  = 0.07.
>
> We plan to add these experiments to the final version of the paper based on the feedback from the reviewer.
>
> >Missing Control/Comparison to Full W′
>
> Thank you for pointing this out. We see that in simple steering, the overall false refusal rates increase model in model vis-a-vis SPECTRA. To see we perform a similar false refusal rate experiment as Table 7 for simple steering find that simple steering increases false refusal rates significantly:
> | Model | Medical | Finance | Law |
> | :--- | :---: | :---: | :---: |
> | **Llama 2** | 8% | 9% | 6% |
> | **Gemma 2** | 5% | 7% | - |
>
> >“Refusal Layer/Token Position Explanation: The layers selected for steering (l_2) are detailed in the appendix (Table 12). The authors should discuss why fine-tuning causes a shift in the refusal layer (e.g., Llama2 Medical: 13/14) and what the mechanistic implication of this shift is.”
>
> Thank you for pointing this out. We believe this occurs due to a change in the basis vectors of each layer during the fine-tuning process hence leading to shift in the position and value of the refusal vector. From the evidence provided in the work, we believe this implicit shift in the position of refusal direction of the language model is correlated with the loss in the safety of the language of the model. Although we believe this claim would need further experimentation and would be a very relevant future work.

---

> ### Author Response · Authors · 2025-11-24
> **Official Comment by Authors (2/2)**
>
> # Questions
>
> >Can the authors provide a direct comparative evaluation between SPECTRA (Eq. 3) and the simpler steering (Eq. 5), specifically showing the coherence retention benefit that \tilde{W} is supposed to provide? Without this direct comparison, the necessity of the low-rank projection, which adds complexity (ActSVD computation, rank selection), is not sufficiently justified.
>
> Thank you for this questions. We have added the relevant experimentation in the response above.
>
> > Why is Variant A (r_1 into W_{l2}^f , i.e., base model refusal vector in fine-tuned model's optimal layer) consistently the most effective? Specifically, why is it superior to Variant C (r_2 into W_{l2}^f), which uses the refusal direction calculated from the fine-tuned model itself? Does this suggest that the fine-tuning process damages the interpretability/saliency of the refusal direction more than it damages the original weights, making the base model's vector a purer signal? Please provide a detailed hypothesis.
>
> We have added a discussion and our hypothesis in the response above.
>
> >Hyperparameter Sensitivity and Generalization: Please provide the sensitivity analysis plots (ASR and coherence metric changes vs. α) for at least two different models/domains (e.g., Llama2 Medical and Gemma2 Finance) to demonstrate that the optimal α and rank p values are stable. How would a practitioner determine optimal α and p without running extensive red teaming evaluations?
>
> We have added these experimentations above. We would also like to add the fact that values of alpha are very small but consistent across various architectures (0.01 - 0.01). We believe our initial experimentation and hyperparameter choices can be a good start when applying our method to other models/domains
>
> >In-Domain Safety Benchmarks: For the Law and Finance domains, the authors had to create custom safety datasets (FinSafeEval, LawSafeEval). Could the authors briefly describe the nature of the harmful prompts in these datasets? For instance, do the Law prompts relate to illegal advice, and the Finance prompts to fraud/manipulation? This context is crucial for understanding the "In-Domain" safety results.
>
> Thank you for pointing this out. We have added the benchmarks we created as supplementary materials. Overall, the content in the benchmark follow the style of unsafe question as in AdvBench, i.e., Law prompts are related to questions on performing illegal activities / civil fraud etc, and finance questions are related to tax fraud/ illicit transactions etc . We chose prompts that best resembled the prompts in domain coherence benchmarks of LegalBench and FinanceBench.
>
> [1]  Wei, Boyi, et al. "Assessing the brittleness of safety alignment via pruning and low-rank modifications." arXiv preprint arXiv:2402.05162 (2024)

---

### Author Response · Authors · 2025-12-04
**Official Comment By Authors**

Based on reviewer feedback, we have made the following changes to paper:
1) Corrected the typos and rewrote section 2 as suggested by Reviewer zaiV.
2) Added comparison to baselines methods, see Appendix A.7, as suggested by reviewer zaiV and 5ZCC. Showing significant impact on false refusal rates of other methods vs ours.
3) Added impact of multi-turn capability as suggested by reviewer hfRS in Appendix A.6.
4) Added the hyperparameter sweep search as suggested by reviewer 5ZCC and hfRS in Appendix A.1.

We would like to thank the reviewers for their review.

---

### Meta-Review · Area_Chair_j836 · 2026-01-05

**Summary:**

While the reviewers see some merits in the proposal of model weight adaptation to adjust the balance between safety (refusal) and general capabilities, this submission should not be accepted in its current form due to several fundamental issues, as pointed out by the reviewers, including

- Limited comparisons to existing baselines - there are many defenses, such as circuit breake,r that provide strong safety by adjusting model weights. Also, comparisons to existing steering vector-based defenses should be included.

- Presentation and clarity need improvements

- It's unclear whether Table 4 presents the results of adaptive attacks or non-adaptive attacks. The fact that TAP has very low ASR appears to be the latter case. The authors should test their defense against adaptive and advanced attacks.

Overall, this paper requires significant modifications and another round of full review.

**Reviewer Concerns:**

Major concerns were not addressed properly, as mentioned in the Summary

**Reviewer Scores:**

Unlikely that any reviewer would increase the score.

---

### Decision · Program_Chairs · 2026-01-26

Reject